# The Influence of Yeast Strain on Whisky New Make Spirit Aroma

Christopher Waymark and Annie E. Hill *

The International Centre for Brewing and Distilling, Heriot-Watt University, Edinburgh EH14 4AS, UK; chriswaymark@yahoo.co.uk
* Correspondence: A.Hill@hw.ac.uk; Tel.: +44-(0)131-451-3458

**Abstract:** Flavour in Scotch malt whisky is a key differentiating factor for consumers and producers alike. Yeast (commonly *Saccharomyces cerevisiae*) metabolites produce a significant amount of this flavour as part of distillery fermentations, as well as ethanol and carbon dioxide. Whilst yeast strains contribute flavour, there is limited information on the relationship between yeast strain and observed flavour profile. In this work, the impact of yeast strain on the aroma profile of new make spirit (freshly distilled, unmatured spirit) was investigated using 24 commercially available active dried yeast strains. The contribution of alcoholic, fruity, sulfury and sweet notes to new make spirit by yeast was confirmed. Generally, distilling strains could be distinguished from brewing and wine strains based on aroma and ester concentrations. However, no statistically significant differences between individual yeast strains could be perceived in the intensity of seven aroma categories typically associated with whisky. Overall, from the yeast strains assessed, it was found that new make spirit produced using yeast strains marketed as 'brewing' strains was preferred in terms of acceptability rating.

**Keywords:** yeast; distillery fermentation; spirit flavour; whisky; new make spirit; aroma

## 1. Introduction

The production of Scotch malt whisky is both beautifully simple and extraordinarily complex. Simplicity lies in the use of only three ingredients and the same basic processing features. However, no two production sites or operational parameters are alike and the myriad choices, from barley variety and malting specification, fermentation regime, still design and operation, through to cask selection and maturation time, lead to immense product diversity. The final flavour and aroma of whisky is due to the interaction of more than 1000 flavour-active compounds termed congeners. The major congener classes that were identified in Scotch whisky are higher alcohols, esters, acids, phenols, lignin-degradation products, and lactones. Studies of processed raw materials, fermentation products, spirit distillate and matured whisky indicate the likely production stage (s) of flavour congeners (Figure 1) and highlight the role of yeast in the production of floral, fruity, and sweet attributes [1].

It is noted that some congener compounds could be generated in multiple stages of the production process. Furfurals and pyrazines from Maillard reactions that provide biscuity, chocolatey or marzipan flavours could arise from conditions in mashing, distillation and cask preparation as well as kilning [2]. Esters are primarily yeast metabolites but are also produced, albeit more slowly, as a result of condensation reactions in the cask [3], or, to a lesser extent, in the still [4]. Phenols are characteristic of peated malt, but also emerge in fermentation [5] and as oak decomposition products in the cask [3]. Lactones are principally formed as oak extractives [3] but were also identified in fermentation [6].

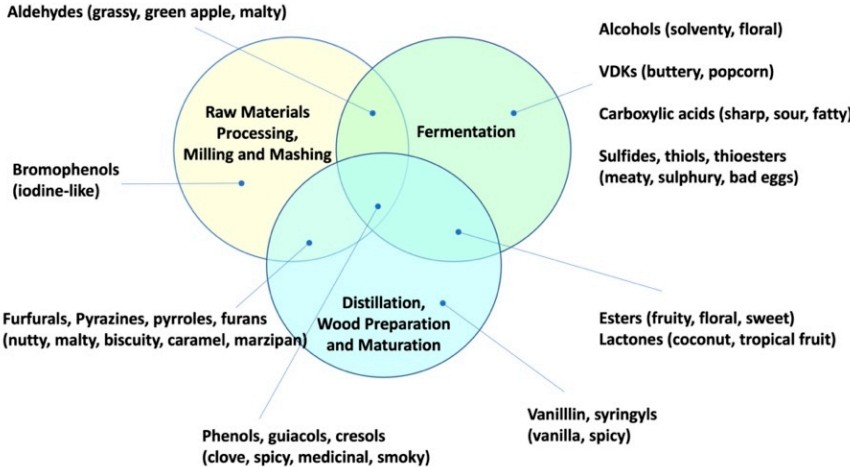

**Figure 1.** Congeners categorized by Scotch whisky production stage (VDKs: Vicinal Diketones) [1].

Under The Scotch Whisky Regulations (2009) [7], for a whisky to be denoted as 'Scotch whisky', it must be distilled and matured in oak casks in Scotland where malted barley and/or other cereal ingredients are processed, converted, and fermented at a Scottish distillery. The stipulation surrounding fermentation is simply that it is 'by yeast'. Yeast strains used in industrial distillery fermentations are predominantly those belonging to *Saccharomyces cerevisiae*. Throughout much of the 20th century, spent brewing yeast was employed in both baking and whisky production, prior to the initial development of specialised leavening yeast strains for baking and subsequent specialised distilling strains with improved characteristics more suited to distillery fermentations. The typical commercial strains that are currently used derive from a hybrid between ale yeast and a wild *S. cerevisiae* strain (previously known as *S. diastaticus*) with amylolytic properties [8]. Such strains are high yielding in terms of ethanol and produce consistent concentrations of desired congeners. As a result, both grain and most malt whisky distilleries predominantly use a sole strain of distilling yeast, with the practice of adding spent brewing yeast having fallen out of favour.

The focus on spirit yield and consistency in congener production is a sensible commercial strategy. However, a number of distillers are seeking a greater diversity in flavour profile in order to either enhance specific attributes or to expand the palette for blending. One route to achieve this is through yeast choice.

In this work, the link between yeast strain selection and new make spirit aroma profile was investigated using 24 commercially available active dried yeast strains. The influence of yeast application type in terms of designation as 'distilling', 'brewing' or 'wine' in predicting new make spirit flavour is discussed.

## 2. Materials and Methods

### 2.1. Wort Preparation

A concentrated batch of unboiled fresh wort (SG of 1.0726) was produced from Distillers Malt (Muntons, Stowmarket, UK) using the International Centre for Brewing and Distilling (ICBD) pilot brewing kit at Heriot-Watt University, Edinburgh. A weight of 55 kg of malt was milled using a two-roller mill (Fraser Agricultural, Inverurie, GBR) then mashed with 150 L of water at 64 °C for 1 h prior to separation using a lauter tun. The grain bed was then sparged continuously with 81 °C water to obtain the desired volume and gravity. Fresh wort was frozen in aliquots and stored at −20 °C. When required, the wort was defrosted and diluted to the desired gravity (1.0612 ± 0.0018). An amount of 850 mL of defrosted unboiled wort was diluted with 150 mL of autoclaved water. Gravities were measured using a benchtop Anton Paar DMA 4500 (Anton Paar, St. Albans, UK).

### 2.2. Yeast Viability

Yeast viability assessments were carried out using the ABER Countstar$^{TM}$ automatic yeast counter (ABER Instruments, Aberystwyth, UK) using methylene blue stain. In a 1.5 mL Eppendorf tube, 10 μL of yeast slurry was pipetted into 990 μL of sodium acetate (0.1 M) (Fisher Scientific, Loughborough, UK) EDTA (10 mM) (Sigma Aldrich, Milwaukee, WI, USA) solution (pH 4.5) and agitated. Into another 1.5 mL Eppendorf tube, 500 μL of the resulting 1000 μL of solution was then transferred into 500 μL of methylene blue staining solution (Fisher Scientific, Loughborough, UK) and agitated. An amount of 20 μL of the stained solution (dilution factor of 200) was then pipetted into a chamber on an ABER Countstar$^{TM}$ slide.

### 2.3. Yeast Rehydration, Pitching and Fermentation

A weight per volume pitching rate of 1 g active dried yeast (ADY) per 1 L of wort was used. For rehydration, ADY was slurried with autoclaved water (30 °C) then placed in an unagitated water bath (Grant SUB36; Grant Instruments, Shepreth, UK) at 30 °C for 60 min prior to pitching.

Fermentations using individual yeast strains (Table 1) were performed. Fermentation time was set at 72 h, and temperature at 35 °C, with fermentations carried out in triplicate for each strain. Fermentations (300 mL) were carried out in 500 mL conical flasks (Fisher Scientific, Loughborough, UK). The triplicate fermentations per yeast strain were consolidated after 72 h to give 900 mL of wash. Wash samples were centrifuged for five minutes (1006.2 g) to remove yeast from suspension (Denley, Guangzhou, China). Wash alcohol by volume (ABV%) calculations were then conducted by first measuring the specific gravity of a centrifuged wash sample using a benchtop Anton Paar DMA 4500 (Anton Paar, St. Albans, UK). The ABV% of the sample was then estimated using the following formula: Alcohol by volume (ABV%) = (Original Gravity − Final Gravity) × 135.25.

### 2.4. Distillation

Wash distillations were carried out using 2 L copper pot stills (operating capacity of 1 L) and associated worm-tub condensers (Copper-Alembic, Gandra, Portugal). Stills were heated using electric hotplates (Duronic, Romford, UK). 900 mL of wash was charged per yeast strain. 500 μL of silicon-based antifoam was added to each 900 mL wash distillation. The collection of low wines was carried out until the distillate coming off the still was 1.5% alcohol by volume (ABV).

Low wines distillations were carried out using 2 L copper pot stills (operating capacity of 1 L) and associated worm-tub condensers (Copper-Alembic, Gandra, Portugal). Approximately 200–300 mL of low wines was charged per yeast strain, and 2.5 mL of foreshots were collected per spirit distillation. After the collection of foreshots, a 3.5 mL sample of distillate (spirit or 'hearts cut') was collected and the ABV (%) recorded. Spirit was collected until the distillate coming off the still was 70% ABV. Feints (distillate produced post spirit cut) were not collected in this experiment.

Alcohol by volume (ABV%) and density values were determined using a handheld Anton Paar DMA35 density meter (Anton Paar, St. Albans, UK).

**Table 1.** Yeast strains used in this study.

| Number | Name | Yeast Strain | Application |
|---|---|---|---|
| 1 | Pinnacle MG+ | *Saccharomyces cerevisiae* | Whisk(e)y |
| 2 | SafSpirits M1 | *S. cerevisiae* | Whisk(e)y |
| 3 | SafSpirits USW6 | *S. cerevisiae* | Bourbon/Whiskey |
| 4 | Pinnacle M | *S. cerevisiae* | Whisk(e)y |
| 5 | Distilamax MW | *S. cerevisiae* | Whisk(e)y |
| 6 | Distilamax GW | *S. cerevisiae* | Whisk(e)y |
| 7 | Pinnacle S | *S. cerevisiae* | Whisk(e)y |
| 8 | Pinnacle G | *S. cerevisiae* | Whisk(e)y |
| 9 | Kerry M | *S. cerevisiae* | Whisk(e)y |
| 10 | Distilamax XP | *S. cerevisiae var. diastaticus* | Whisk(e)y |
| 11 | Safale T58 | *S. cerevisiae* | English/Belgian ale |
| 12 | Safale WB06 | *S. cerevisiae* | Wheat beer |
| 13 | Safale BE-256 | *S. cerevisiae* | Belgian ale |
| 14 | Safale S-04 | *S. cerevisiae* | US/English ale |
| 15 | Saflager 189 | *S. pastorianus* | Swiss lager |
| 16 | Safale K-97 | *S. cerevisiae* | German/Belgian ale |
| 17 | Safale S-33 | *S. cerevisiae* | Belgian/English ale |
| 18 | Safale BE134 | *S. cerevisiae var diastaticus* | Belgian-Saison |
| 19 | Hothead | *S. cerevisiae* | Norwegian ale |
| 20 | Kveik | *S. cerevisiae* | Norwegian ale |
| 21 | LalvinV116 | *S. cerevisiae* | Ice wine |
| 22 | Lalvin EC1118 | *S. cerevisiae var. bayanus* | Champagne |
| 23 | Lalvin ICV OKAY | *S. cerevisiae* | Wine |
| 24 | Exotics SPH | *Hybrid S. cerevisiae/S. paradoxus* | Red wine |

## 2.5. Gas Chromatography Mass Spectrometry (GC/MS)

18 new make spirit (NMS) samples were analysed to determine specific ester concentrations using a solid-phase microextraction (SPME) GC/MS method developed within Heriot-Watt University's International Centre for Brewing and Distilling (ICBD). Specific ester concentrations were determined using a Shimadzu QP2010 Ultra (Shimadzu, Kyoto, Japan) GC/MS integrated system with split/splitless injector and Shimadzu AOC 5000 autosampler. SPME was conducted using a 65 mm polydimethylsiloxane/divinylbenzene coated fibre (Supleco, Bellefonte, PN, USA). Samples were prepared by adding 1 mL of NMS to a 15 mL vial and 5 mL of distilled water then added to the NMS and mixed. Data were handled using Shimadzu GC/MS Solutions data handling systems. For instrument details see Table 2.

**Table 2.** GC/MS Instrument conditions.

| | |
|---|---|
| SPME | Conditioning: 5 min at 70 °C |
| | Extraction: 5 min |
| | Desorption: 1 min |
| Column | DB Wax UI (30 m × 0.25 mm × 0.25 μm) (Agilent, Santa Clara, CA, USA) |
| Carrier gas | Helium (BOC; 5.0) |
| Internal standard | Methyl heptanoate |
| Oven | 40 °C for 3 min; ramp to 100 °C at 10 °C/min; ramp to 160 °C at 4 °C/min; ramp to 220 °C at 10 °C/min. Hold for 100 min. Ramp to 250 °C at 70 °C/min. Hold for 3 min. |

## 2.6. Sensory Analysis

Quantitative descriptive analysis: Aroma Intensity Rating. A form of quantitative descriptive analysis (QDA) was carried out on the NMS samples. Sensory analysis was carried out in two phases: (1) Language familiarisation, (2) NMS nosing, undiluted (still strength).

The language familiarisation sessions utilised descriptors from the Scotch whisky research institute (SWRI) whisky wheel [9] and aroma standards using the Whisky aroma

kit (The Aroma Academy, Aberdeen, UK). A total of 200 min of language familiarisation was carried out with the panel (*n* = 5).

To avoid the introduction of bias derived from numerical scales, the NMS nosing ballot constructed for this experiment was an adaptation of a graphic line scale [10]. In place of a line scale it consists of three tick boxes, each with a corresponding word anchor relating to the intensity of the descriptor. The word anchors were, from left to right: 'Not present', 'Mild/Subtle' and 'Strongly present'. Unbeknownst to the assessors, the tick boxes represented a scale of 0–1.0. The assessor responses were converted to their numerical value by the panel leader after the conclusion of the sensory session.

### 2.7. Hedonic Assessment: Acceptability Rating

An acceptability test using the nine-point hedonic scale was also performed. The nine-point hedonic scale was historically used for the hedonic assessment of food, beverages and non-food products [11].

Assessors (*n* = 5) were asked to nose samples individually and then check only one of the phrases from the nine presented on a ballot that best described their opinion of the sample. The phrases represented a scale of 1–9 where 9 represented 'like extremely' [11]. This assessment was performed in order to narrow down the initial selection of yeast strains [12].

### 2.8. Statistical Analysis

Kolmogorov–Smirnov, Analysis of Variance (ANOVA), Friedman and Games Howell multiple comparison tests were performed using the IBM SPSS statistics software package (IBM, Armonk, NY, USA).

The Kolmogorov–Smirnov test compares an observed sample distribution and a specified theoretical distribution and is sometimes referred to as a normality test [13]. Analysis of variance (ANOVA) is a statistical technique that tests to determine if significant differences exist between three or more groups and this includes one group with multiple variations [14] such as the yeast used in this work. The Friedman test in a non-parametric one-way ANOVA [15].

The Games Howell multiple comparison test is a pairwise comparison procedure, designed for use with unequal sample sizes, that can determine which specific groups are statistically significantly different [16,17].

### 3. Results

### 3.1. Yeast Viability and Wash Alcohol Volume

The yeast viability, measured using methylene blue staining following 60 min of rehydration in water at 30 °C, was found to be less than 90% for all strains tested. However, no significant difference was found in the average final-wash alcohol by volume (ABV). Ten of the yeast strains assessed achieved average final-wash ethanol concentrations of 8% ABV or over. This included all of the yeast strains marketed as 'distilling' strains (1–10), with the exception of yeast strain 7 (Pinnacle S).

### 3.2. Aroma Intensity Ratings

The aroma intensity ratings experiment intended to investigate whether the yeast strain used to produce the new make spirit had an effect on the perceived intensity of seven common aromas associated with whisky new make spirit. Seven individual Friedman tests were carried out.

From the individual Friedman tests, it was found that there was no statistically significant difference in the perceived intensity of the fruity (fresh) aroma depending on the yeast strain used to produce the new make spirit ($\chi^2$ (22) = 25.26, $p$ = 0.285). This was also the case for all of the other aroma categories; fruity (2), sulphury, floral, sweet, cereal and feinty ($\chi^2$ (22) = 27.03, $p$ = 0.210, $\chi^2$ = 27.49, $p$ = 0.193, $\chi^2$ = 13.75, $p$ = 0.910, $\chi^2$ = 31.63, $p$ = 0.084, $\chi^2$ = 25.14, $p$ = 0.290 and $\chi^2$ = 30.31, $p$ = 0.11, respectively).

Distilling yeast strain 2 achieved the highest average intensity rating for the fruity (fresh) aroma category (0.7; Figure 2a), with both distilling and wine yeast receiving an average score of 0.45.

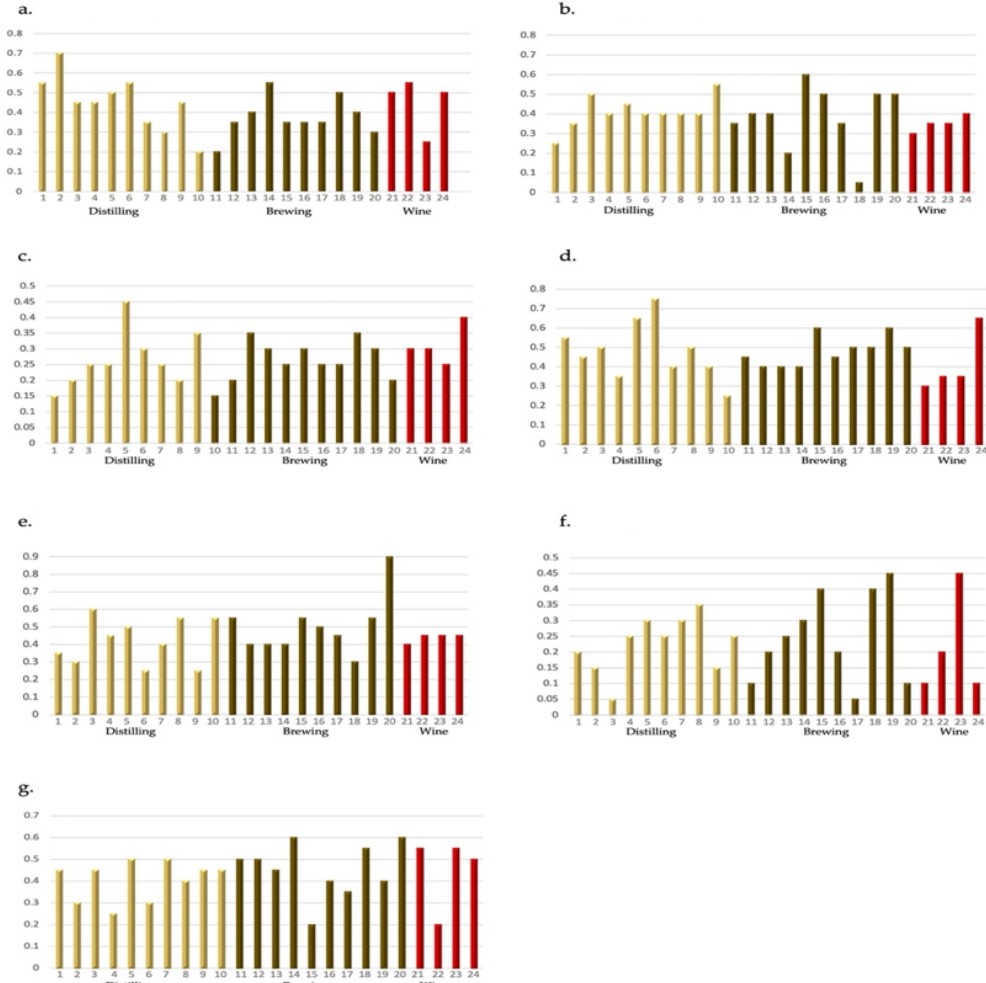

**Figure 2.** Average aroma intensity rating from 0 (absent) to 1.0 (most intense) for seven common aromas associated with whisky new make spirit: (**a**) Fruity (Fresh), (**b**) Fruity (Dried), (**c**) Floral, (**d**) Sweet, (**e**) Cereal, (**f**) Sulfury, (**g**) Feinty. Ratings are grouped by yeast type: Distilling, Brewing, or Wine; *n* = 5.

This suggests that the NMS spirit produced using these strains possesses the fruity characteristics of the solvent-like and fresh fruit variety. As described in the Scotch Whisky Research Institute (SWRI) flavour wheel [9].

The 'Fruity (dried)' category relates to the dried fruit aroma category described on the SWRI whisky flavour wheel [9]. Brewing yeast strain 18 achieved the lowest average rating for this attribute (0.05) (Figure 2b). Regardless of classification, the average intensity rating was below 0.5 in relation to the Fruity (dried) attribute. This suggests an overall lower perception of this attribute by the panel in the given samples. In addition, only NMS samples produced using distilling yeast 10 and brewing yeast 15 could be considered as exhibiting mild-to-strong fruity characteristics of the dried fruit variety (Figure 2b).

In relation to the floral attribute, which is associated with fresh flower-like aromas, all strains achieved average intensity ratings below 0.5 (Figure 2c). More specifically, with the exception of yeast strain 5 (0.45), all of the new make spirit produced achieved average aroma intensity ratings or 0.4 or lower, which suggests that the NMS produced using the yeast strains included in this study possessed weak floral aroma characteristics.

The sweet aroma category was associated with honey, vanilla and caramel-like aroma [9]. Six yeast strains achieved average aroma intensity ratings above 0.5, with yeast strain 6 achieving the highest average rating of 0.75 (Figure 2d). With the exceptions of yeast strains 4, 10, 21, 22 and 23, all of the assessed yeast strains achieved average intensity ratings of 0.4 or above (Figure 2d) suggesting that the NMS produced using these yeast strains exhibits mild-to-strong sweet aroma characteristics. Interestingly, three of the four wine yeast strains achieved scores below 0.4, with the average score of 0.41 caused by a single higher-scoring strain. Overall, distilling and brewing strains were ranked higher with an average score of 0.48.

Brewing yeast strain 20 scored significantly high in relation to cereal notes (0.9; Figure 2e), which include toasted, malt and husky aromas. On average the brewing yeast scored higher (0.5) than both the distilling and wine yeast (0.42 and 0.43, respectively).

Yeast strains 3, 11, 17, 20, 21 and 24, achieved average intensity ratings of 0.1 or lower in relation to the sulfury aroma category. This suggests a particularly low perception of this attribute by the assessors in these NMS samples (Figure 2f). Yeast strains 19 and 23 on the other hand both achieved the highest average rating for this attribute at 0.45 (Figure 2f). Overall, the mean average intensity ratings did not exceed 0.3 across all of the yeast strains assessed suggesting that sulfury characteristics in the NMS were minimal.

The average aroma intensity rating for feinty notes, which include leathery, tobacco and sweaty aromas, was consistent across all yeast strains with brewing and wine yeast averaging 0.45 and distilling yeast 0.4. Of the brewing yeast, strains 14 and 20 were the highest ranking.

### 3.3. Ester Profiles

The concentration of seven esters was assessed in new make spirit samples from eight distilling yeast strains, eight brewing yeast strains and two wine yeast strains used in the aroma intensity test. Ethyl hexadecanoate was not detected in any of the samples. The results for ethyl hexanoate, ethyl lactate, ethyl decanoate, phenylethyl acetate, ethyl dodecanoate and ethyl octanoate are shown in Table 3.

**Table 3.** Average ester concentration in new make spirit per yeast type: Distilling (*n* = 8), Brewing (*n* = 8), Wine (*n* = 2).

| Ester | Aroma | Aroma Threshold (mg/L) [18] | Average Concentration per Yeast Type (mg/L) | | |
| --- | --- | --- | --- | --- | --- |
| | | | Distilling | Brewing | Wine |
| Ethyl hexanoate | Aniseed, apple-like | 0.076 | 1.71 | 0.52 | 1.60 |
| Ethyl lactate | Buttery, butterscotch, artificial strawberry | >14 | 1.30 | 23.90 | 7.97 |
| Ethyl decanoate | Floral, soapy | 1.1 | 1.09 | 0.79 | 1.96 |
| Phenylethyl acetate | Roses, honey, waxy | 0.7 | 1.63 | 0.92 | 2.47 |
| Ethyl dodecanoate | Soapy, estery | 0.64 | 0.25 | 0.56 | 0.92 |
| Ethyl octanoate | Sour apple | 0.24 | 0.75 | 0.71 | 1.41 |
| | | Total | 6.73 | 27.4 | 16.33 |

All samples exceeded the threshold value for ethyl hexanoate with distilling yeast generating new make spirit with the highest concentration. Samples from brewing yeast contained an average of 23.90 mg/L ethyl lactate, exceeding the aroma threshold of 14 mg/L.

Samples from wine yeast contained the highest concentration of ethyl decanoate, phenylethyl acetate, ethyl dodecanoate, and ethyl octanoate, with each case exceeding threshold levels. Threshold levels were not reached for either distilling or brewing strains for ethyl decanoate and ethyl dodecanoate.

The total ester concentration, by addition of each of the six individual averages, places brewing strains highest followed by wine and then distilling yeast (Table 3).

### 3.4. Hedonic Assessment: Acceptability Ratings

New make spirit samples produced using the 24 different yeast strains used in this study were assessed individually and assigned an acceptability rating using a ballot constructed based on the 9-point hedonic scale.

It is generally not recommended to perform statistical analyses of data derived from hedonic assessments such as the 9-point hedonic scale, as the type of data collected is not mathematically suitable. However, mean ratings can be collected and ratings of seven or higher are considered indicative of a product of a highly acceptable sensory quality [19].

As shown in Table 4, seven of the twenty-four yeast strains achieved an average acceptability rating of 7 or above.

**Table 4.** Average mean acceptability ratings of new make spirit samples that achieved a mean rating of 7 or above (out of 9).

| Yeast Strain | Mean Acceptability Rating |
| --- | --- |
| 14 | 7.75 |
| 2 | 7.5 |
| 13 | 7.5 |
| 20 | 7.5 |
| 11 | 7.25 |
| 16 | 7 |
| 17 | 7 |

Yeast strain 14 (brewing yeast) achieved the highest average acceptability rating. Yeast strains 2, 13 and 20 (distilling, brewing, and brewing yeast, respectively) jointly achieved the second highest average acceptability rating (Table 4). Thirdly, these were followed by yeast strain 11 (brewing yeast) and then joint fourth by yeast strains 16 and 17 (both brewing yeast).

From these data and the selection of yeast strains used in this study, it appears that, with the exception of spirit produced using yeast strain 2, NMS produced using 'brewing' strains were found to be the more acceptable products by the panel.

## 4. Discussion

Multiple studies identified that different yeast strains produce different congener profiles and different aromas in beer [20–22]. Furthermore, the mechanisms for production of higher alcohols and esters were well studied [23,24]. The difference in congener profiles between different strains, (even within species), depends on differences in the metabolism of sugars, proteins and amino acids and autolysis of yeast [25].

Overall, although statistical analysis produced no credible evidence from this study that the yeast strain had a statistically significant impact on the perceived intensity of the aroma categories assessed, it is suggested from the results of the panel responses that there were differences in the overall perceived aroma profiles of the assessed new make spirit (NMS) profiles and that a large selection of the yeast strains used in this study produced NMS with distinctly fresh-fruit-like, sweet and non-sulphury characters.

Commercially, yeasts are commonly split and marketed into groups related to their intended industrial application. To look more broadly at the yeast strains involved in this study, the strains were separated into three groups: distilling strains, brewing strains, or wine strains, based on either the industrial application or fermentation medium specified by the yeast manufacturer. The categorisation of strains based on application (distilling, brewing, or wine) highlighted distinct patterns. Generally, distilling yeast was regarded as fruitier than brewing or wine yeast. Brewing yeast was regarded as having cereal, sulfury, and feinty notes, and wine yeast was rated alongside distilling yeast for fresh fruit aroma and highest for floral aroma (although intensity ratings for floral aroma were low across all samples). These results compare with analytical studies of ester concentration, with wine yeast containing, on average, higher concentrations of ethyl decanoate, phenylethyl acetate, ethyl dodecanoate, and ethyl octanoate, compared to distilling and brewing yeast.

The aroma characteristics of these esters are floral, rose, estery, and sour apple, respectively. These results also concur with data from the literature and unpublished industry studies; higher alcohols, acetate esters and ethyl esters are more commonly cited in fermentation studies with distilling yeast in comparison with brewing or wine yeasts [1,12].

That distilling yeast was perceived as fruitier than brewing or wine yeast contrasts with the total ester concentration (Table 3). However, the analysis of aroma compounds and the perceived character of matured Scotch whisky demonstrates that the relative levels of individual esters within the headspace may be low but the interactions with other components within the spirit matrix leads to an increase in the overall 'estery' character [26].

The results from the hedonic assessment clearly indicate that brewing strains produce new make spirit that is 'most acceptable'. Six of the top seven scoring strains were brewing strains, with each strain also ranking highly in at least one attribute within the aroma intensity ratings. On average brewing strains also produced a spirit with a significantly higher concentration of ethyl lactate, a compound associated with buttery, butterscotch and artificial strawberry characteristics. Traditionally, brewing strains were employed within distillery fermentations either as the main production yeast, or in co-fermentation at additions of up to 50%. The results from this study highlight the positive influence that brewing strains of *S. cerevisiae* have on new make spirit aroma.

Borneman et al. [27] warns that high levels of genetic similarity exist between commercial *S. cerevisiae* strains which potentially reduce the scope for genetic, metabolic, and thus new congeneric differences. The lack of statistically significant differences between individual strains in both aroma intensity ratings and in ester production in this study reflects this theory. Lambrechts and Pretorius [28] reviewed the production of higher alcohols and esters in *Saccharomyces* strains and found them to be strain-dependent. For alcohols this is potentially due to differences in amino acid utilisation in the Ehrlich pathway. These metabolic differences were found to be greater in non-*Saccharomyces* yeasts [29]. For ester synthesis, the precursor alcohol and acyl-CoA concentrations, as well as the expression of the Eeb1 and Eht1 genes [30], were shown to be key [31]. Saerens et al. [32] later reported that yeasts could be chosen for the production of desired alcohols and esters based on the expression levels of genes involved in their biosynthesis. The production of congener types by non-conventional yeasts, such as *Hanseniaspora guillermondii,* is cited commonly in the literature, with multiple studies demonstrating higher concentrations of higher alcohols and esters in comparison to brewing and wine yeast strains [33,34]. As such, there is a great potential in their application within distillery fermentations.

## 5. Conclusions

Developments in yeast isolation, propagation techniques, and format have greatly expanded the diversity of the yeast strains available to Scotch whisky distillers. It is also no secret in the distilling industry that using alternative yeast strains, compared to the 'industry standards', is a perfectly viable production strategy. There are already a number of products available on the Scotch whisky market that are produced with yeast strains that are either proprietary or strains that were not initially developed and marketed for whisky fermentations.

This work highlights the aroma characteristics imparted by yeast, the variation in overall aroma perception, and consequently the importance of yeast strain selection. There is clear scope to increase the palette of flavour attributes in new make spirit giving distillers an opportunity to innovate in this very traditional process.

**Author Contributions:** Conceptualization, A.E.H.; methodology, A.E.H.; investigation, C.W.; writing—original draft preparation, A.E.H.; writing—review and editing, C.W.; supervision, A.E.H.; project administration, A.E.H.; funding acquisition, A.E.H. All authors have read and agreed to the published version of the manuscript.

**Funding:** This research was part-funded by Innovate UK and The Port of Leith Distillery.

**Institutional Review Board Statement:** The study was conducted according to the guidelines of Heriot-Watt University and approved by the Ethics Committee of the School of Engineering and Physical Sciences (approval number 19/EA/AH/1, 9 July 2016).

**Informed Consent Statement:** Informed consent was obtained from all subjects involved in the study.

**Data Availability Statement:** The data presented in this study are available on request from the corresponding author. The data are not publicly available due to commercial sensitivity.

**Acknowledgments:** Many thanks to Victoria Muir-Taylor for practical work and Maarten Gorseling for analytical support.

**Conflicts of Interest:** The authors declare no conflict of interest.

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
