# Peer review of "The Influence of Yeast Strain on Whisky New Make Spirit Aroma"

_fermentation, doi:10.3390/fermentation7040311_

Round 1
Reviewer 1 Report
This is a generally well-written manuscript on the influence of selected yeast strains on whisky spirit aroma characteristics. The subject matter should be of interest to the readership of the Special Issue.
Specific comments/suggested revisions
- Abstract: line 10 - change to .....there is limited information on the relationship....
- Need to define "new make spirit" for readers unfamiliar with whisky production. For example, change line 11 to ....new make spirit (freshly distilled, un-matured spirit).... Also past tense needed (was investigated)
- line 12 - ....dried yeast strains....
- line 12 - fruity
- line 13 - "On average" is confusing. Perhaps: Generally, .....
Introduction
- line 70 - past tense (was investigated)....
- line 71 - instead of yeast "class", use yeast application type
Materials & Methods
- Need more information on choice of yeast strains employed in this study. Suggest a Table listing species names (some may be Saccharomyces pastorianus), application (ale, lager, distilled spirit, type of wine), source, commercial name (if any)
- line 89 - no need for capital letters for sodium acetate
- line 96 and elsewhere - need correct information on fermentation conditions (pitching rate? media volume? temperature?)
- Model of Anton Paar meter? (line 125)
- Sensory panel (line 149) - were they experienced/trained? only 5 panellists?
- line 155 - past tense (were converted)
- line 161 - number of Assessors?
Results
- Table 2 indicates only 18 yeasts, but 24 are indicated elsewhere in manuscript
Discussion
- line 288 - need to spell out NMS when first used
- line 296 - Instead of "classification" (which has a distinct meaning for yeast taxonomists, suggest categorisation
- line 297 - Delete "On average", replace with Generally, ....
- line 334 - not an appropriate reference to cite for primary literature on use of non-conventional yeasts
Conclusions
- Line 338-339 - This is not a true statement!
- If brewing yeasts provided more flavoursome NMS, what about their overall fermentation performance/ethanol yields when fermenting malt wort for whisky production, compared with existing distilling strains? Some discussion merited.
Author Response
Many thanks for the comments/suggestions for the paper. The manuscript has been updated to reflect these as follows:
Specific comments/suggested revisions
- Abstract: line 10 - change to .....there is limited information on the relationship.... – corrected.
- Need to define "new make spirit" for readers unfamiliar with whisky production. For example, change line 11 to ....new make spirit (freshly distilled, un-matured spirit).... Also past tense needed (was investigated) - corrected.
- line 12 - ....dried yeast strains.... - corrected.
- line 12 – fruity - corrected.
- line 13 - "On average" is confusing. Perhaps: Generally, ..... - corrected.
Introduction
- line 70 - past tense (was investigated).... - corrected.
- line 71 - instead of yeast "class", use yeast application type - corrected.
Materials & Methods
- Need more information on choice of yeast strains employed in this study. Suggest a Table listing species names (some may be Saccharomyces pastorianus), application (ale, lager, distilled spirit, type of wine), source, commercial name (if any) – Table added.
- line 89 - no need for capital letters for sodium acetate - corrected.
- line 96 and elsewhere - need correct information on fermentation conditions (pitching rate? media volume? temperature?) – further information added to materials and methods.
- Model of Anton Paar meter? (line 125) – added.
- Sensory panel (line 149) - were they experienced/trained? only 5 panellists? – Panellists received 200 minutes of specific training (stated in methods).
- line 155 - past tense (were converted) - corrected.
- line 161 - number of Assessors? – added.
Results
- Table 2 indicates only 18 yeasts, but 24 are indicated elsewhere in manuscript – analysis was carried out on 18 of the 24 yeast (stated in methods).
Discussion
- line 288 - need to spell out NMS when first used – added.
- line 296 - Instead of "classification" (which has a distinct meaning for yeast taxonomists, suggest categorisation – changed.
- line 297 - Delete "On average", replace with Generally, .... – added.
- line 334 - not an appropriate reference to cite for primary literature on use of non-conventional yeasts – two original research papers added to replace ref [1].
Conclusions
- Line 338-339 - This is not a true statement! – ‘yeast format’ has been added to this sentence. It relates to the number of commercially available strains that are now available from yeast supply companies. There are also a number of yeast suppliers that are working with distilleries to isolate or develop bespoke strains.
- If brewing yeasts provided more flavoursome NMS, what about their overall fermentation performance/ethanol yields when fermenting malt wort for whisky production, compared with existing distilling strains? Some discussion merited. A comment on the average %abv has been added to the results. Fermentation profiles were comparable across all strains tested.
Reviewer 2 Report
Dear authors and editors,
The authors present an interesting study regarding the production of new make spirits using yeast strains. The scope of the article is to establish a relationship between yeast strain and observed flavour profile. The paper presents an analysis of the content of esters of new products. Also, the results are correlated with the rate of acceptability and aroma intensity, performed by the specialized assessors.
In order to raise the quality of the paper, my comments are as follow:
1. The authors should mention the names of the 24 yeast strains used in the study.
2. Yeast viability should be discussed in the paper. The pH of obtained products should be correlated with the viability of yeasts and the chemical contents of the final beverages.
3. The data from the panel assessors should be correlated with the chemical content and type of the yeasts analyzed.
4. The authors should analyze also the content of aldehydes and N derivate that have an important role in the aroma of the spirits.
5. A discussion revealing the content of higher alcohols is also necessary.
6. The rate of acceptability should be correlated with the content of S and N derivate.
7. The references [25] reviewed the production of higher alcohols and esters in Saccharomyces strains and found them to be strain-dependent. The authors should analyze the content of higher alcohols in order to correlate the results with this reference.
8. The references [24] and [25] from the manuscript does not correspond with their numbers in the Bibliography.
9. The references [28] is useless as no discussion about the genes and their role in the production of distilled beverages is in the paper.
10. The conclusions should be more specific regarding the results.
11. The graphics should use colours for good visualisation of data.
Author Response
Many thanks for the comments/suggestions for the paper. The manuscript has been updated to reflect these as follows:
In order to raise the quality of the paper, my comments are as follow:
1. The authors should mention the names of the 24 yeast strains used in the study. A table listing the strains has been added.
2. Yeast viability should be discussed in the paper. The pH of obtained products should be correlated with the viability of yeasts and the chemical contents of the final beverages. A comment on viability has been added to the results. Further detail could be added if needed.
3. The data from the panel assessors should be correlated with the chemical content and type of the yeasts analyzed.
4. The authors should analyze also the content of aldehydes and N derivate that have an important role in the aroma of the spirits.
5. A discussion revealing the content of higher alcohols is also necessary.
6. The rate of acceptability should be correlated with the content of S and N derivate.
7. The references [25] reviewed the production of higher alcohols and esters in Saccharomycesstrains and found them to be strain-dependent. The authors should analyze the content of higher alcohols in order to correlate the results with this reference.
Comments 2-7: The focus of the research was on ester production due to a desire to find yeast strains producing increased fruity/estery profile. GCxGC data has been generated for the ‘top 7’ yeast strains in the hedonic assessment which could be included to enable discussion of chemical content for NMS produced by these strains, but as this was not carried out on all 24 it seems out with the scope of this paper.
The references [24] and [25] from the manuscript does not correspond with their numbers in the Bibliography. Corrected (additional line break removed).
9. The references [28] is useless as no discussion about the genes and their role in the production of distilled beverages is in the paper. This is included to enable further reading.
10. The conclusions should be more specific regarding the results. An additional line has been added. In order to reduce repetition, parts could be replaced, if desired.- The graphics should use colours for good visualisation of data. Thank you – colour figures have been added (Figures 1 and 2).
Round 2
Reviewer 2 Report
Dear editors,
I have examined the manuscript carefully. The authors carried out modifications in the abstract, introduction, discussion, conclusions and references parts. In general, the authors received the comments and suggestions made by the evaluators.
Taking into account this new version, I recommend this manuscript for publication.